# Leveraging Quasi-Experimental Methods to Estimate Model Structure: Understanding School Funding Changes in Response to Court Orders

**Kawika Pierson** *[ID] and **Jon C. Thompson**

Atkinson Graduate School of Management, Willamette University, 900 State Street, Salem Oregon, 97301, OR 503-370-6238, USA; jcthomps@willamette.edu

* Correspondence: kpierson@willamette.edu

**Abstract:** We discuss an approach to using the event study, a common experimental design in the social sciences, to parameterize delays and develop other insights into system structure. We show a step-by-step process for undertaking a delay event study, discuss some of the conceptual reasons that this provides information about the delay, and illustrate the process for a typical example. We find evidence that school funding changes following court orders do not adjust quickly, and likely follow a higher-order, as opposed to a first-order, delay process. Our tests also suggest that school district budget makers appear to forecast revenue pessimistically, contributing an additional source of delay to the system.

**Keywords:** delay estimation; event study; delay order; delay parameterization

---

## 1. Introduction

The consensus among scholars of system dynamics for the past 40 years [1] has been that, while it may be possible to use simple statistical models for the estimation of delay characteristics, delays are best understood through the application of more complicated methods [2], or as one degree of freedom in a larger model calibration exercise [3]. This position is most influentially espoused by Sterman [4] (p. 437), who cites multicollinearity and high data requirements to quickly dismiss the feasibility of parameterizing a delay using ordinary least squares.

This critique no longer seems as convincing as it once may have been. First, most modern system dynamics work takes place within a sea of increasingly rich data on the relationships of interest. Rather than abandoning a tool that has high data needs, it would be far better to delineate exactly what the data requirements of the tool are so that modelers can know whether they are able to use it. While the delay parameterization tool we document here does require a certain data structure, the simple fact is that high data requirements are not as troublesome as they might have been 20 years ago.

Second, multicollinearity inflates the standard errors in a linear regression, but it does not inherently bias the slope estimates [5]. When the goal is insight into a system's structure rather than causal inference, even statistically insignificant parameter estimates can be very informative, and so the potential for multicollinearity on its own is not a good reason to abandon a method. Additionally, the test that we illustrate here, which we call a delay event study, can plausibly control for a good deal of the covariation between similar time series by incorporating a collection of fixed effects along with the parameters of interest.

When you consider the potential of regression tests to enable rapid, straightforward insight into the structure of complicated social systems without the need for any model structure to be formulated, it seems limiting to argue that large data requirements and the need for careful setup and interpretation

should disqualify these tools. Yet, despite the common refrain that the optimal situation for any SD model is one where all of the model parameters can be estimated "below the level of aggregation of the model" [6], no work in system dynamics has documented a process for using linear regression to estimate the structure of a system, or demonstrated an example of this process working in practice. We think that this omission has caused some system dynamics scholars to skip over these methods in favor of more impressive sounding tools, or worse to ignore delay parameterization and order estimation altogether during model formulation.

This paper documents one approach to delay parameterization that mimics the quasi-experimental designs that have become increasingly popular among social scientists [7,8]. We call this approach a "delay event study". This method relies on having data for multiple examples of a system of interest (for instance, multiple businesses, school districts, or employees), each observed more frequently than the expected delay length. The second data requirement is that we can identify an event that will shock some subset of the system examples, causing them to adjust through the delay we are interested in parametrizing. The shock can be external or internal to the system and can occur at different times for different observations or at the same time for all of them. The only requirement is that we expect the event to cause a change in the relationship we are estimating the delay characteristics of. The delay event study directly estimates the response of the variable to the event, with finer grained estimates available as the frequency of data measurement increases.

We apply this approach to better understand a specific delay in a complex policy process: The delay between when a judge orders a state to change its school district funding and the time when those funding changes occur. This delay is central to the interpretation of a large body of economics scholarship interested in school funding [9,10], because these papers leverage a series of court orders known as the "adequacy era" that required states to provide adequate education to their citizens, as exogenous instruments in tests that estimate how effective increases in school spending are at improving educational outcomes. Unfortunately, the statistical models used by most of these papers implicitly assume that the court order and the change in funding are contemporaneous. We apply a delay event study to carefully analyze whether this assumption is warranted. Did districts change spending quickly, or did school spending respond only with a significant delay?

The answer is unlikely to change the overall nature of the results economists have found, since school spending has had decades to influence student outcomes by now. However, it can certainly help to cut through the confusion caused by the inconsistency of many early results [11], and could also provide context to more recent null results in settings where insufficient time has passed to begin to see real spending changes or impacts (for instance [12]). More broadly though, this identification strategy is very common across the social sciences, and so our results also comment on the larger issue of how the common mental model, which assumes that the causes and effects of policies are collocated in time, is limiting the collective ability of social scientists to reach correct conclusions in quasi-experimental settings.

That said, this paper is written primarily for system dynamists, not for economists or social scientists. In this context, our paper serves as a reminder that we do not always need to reach for the most complicated tools when we are building confidence in our models. We derive very rich insights into the structure of this complex system using a simple series of distributed lag regressions, and as a result, it is possible that other modelers will have similar experiences in diverse settings. It would be interesting to learn how well or poorly this tool performs in other contexts, with the hope that system dynamics modelers may soon know far more about when they can safely use a linear regression to estimate the dynamics of a delay, and when they will need to reach for more sophisticated tools.

## 2. Performing a Delay Event Study

Simply, a delay event study attempts to detect the response of the variable of interest to a shock. If you have a single example of a complex system and you know that an event occurred at time *t*, which

will eventually cause variable $X$ to change, then figuring out the length of the delay between the event and the change in $X$ might be as simple as very carefully examining a graph of the time behavior of $X$.

In any realistic setting however, this approach would not be easy to apply. It would be very time consuming to run a visual test across many observations and it might also be inconclusive, since there is likely to be heterogeneity in both the perception of and reaction to the event across cases. Worse, any inconsistencies in when the event occurs for different cases might require us to disentangle the event's "signal" from the "noise" driven by the underlying pattern of behavior of the systems. This would be essentially impossible to do visually.

Luckily, many decades of work by economists, starting with Granger's [7] work on causal inference, has honed a class of regression models that accomplish exactly this task. Equation (1) documents one popular formulation:

$$Ln(X_{it}) = \alpha + \sum_{\tau=0}^{T} \beta_\tau D_\tau + \delta_i + \delta_t + \delta_g + \theta \cdot Y_{it} + \varepsilon_{it} \tag{1}$$

where $i$ indexes observation units while $t$ indexes observation times. $X$ is the variable we expect to change as a result of the event, $\alpha$ is the constant of the regression, $\beta$ is a vector of slope estimates, and $\varepsilon$ is the error term. Call the sampling frequency of the data $\Delta t$. If we define a time $\tau$ when the event[1] occurs for each observation $i$, then the sum in Equation (1) is composed of a series of variables $D_\tau$ which depend on $\tau$. $D_0$ takes a value of 1 only when $t$ equals $\tau$, and takes a value of 0 for all other $t$. $D_1$ takes a value of 1 when $t$ is one $\Delta t$ larger than $\tau$, $D_2$ equals 1 when $t = \tau + 2\Delta t$, and so on. The final $D_T$ should be constructed so that it takes a value of 1 for all time t greater than or equal to $\tau + T\Delta t$. Make sure to include enough $D_\tau$ so that the entire delay response can be observed in the output of the test[2].

The two variables $\delta_i$ and $\delta_t$ are observation-based and time-based fixed effects, which will help to control for the way that each individual system might experience a different average level of $X$, or the way that different moments in time might be associated with different levels of $X$ on average. Fixed effects also take values of 1 or 0 but do so either for a specific observation or for a specific time (for a full explanation see [13]). An individual level time trend $\delta_g$ is also included [14]. This control allows each unit of analysis to exhibit its own characteristic pattern of exponential growth or decay for the variable of interest over time. Together, these fixed effect controls combine to allow the regression to differentiate between changes in response to the event and changes caused by many outside influences on the variable $X$.

$Y$ in Equation (1) stands in for a set of control variables that can optionally be added to the event study, and $\theta$ is a vector of slopes for those controls. When selecting controls, it is important to focus only on variables that change the way that observation units will respond to the event. It is not important to control for all the influences on $X$ that are absent from the model if those omissions do not impact the response of each unit to the event.

Estimate Equation (1) using your favorite statistical package[3]. Since Equation (1) uses the natural log of $X$, the resulting slope estimates $\beta_0 - \beta_T$ can be interpreted as giving the percentage change in $X$ at each of the $T$ times following the event[4]. Graphing these estimates provides you with a picture of

---

1　　$\tau$ must be aligned with the data such that all $\tau$ are an integer multiple of $\Delta t$, but it can potentially be different for each $i$. If the event does not occur for some observation units simply set all $D_\tau$ for those observations to zero. This will enable that observation to serve as a control.

2　　That is, make sure that $\Delta t^* D_\tau$ is about twice as large as the delay time you expect to observe.

3　　If you have a very large number of observation units you might consider a package such as Stata's reghdfe [15] that will greatly reduce the computational time needed to arrive at the estimates.

4　　The actual relationship is that $e$ raised to the power $\beta$ will return 1 plus the percentage change for relatively small values of $\beta$.

the average response of $X$ following the event[5], potentially revealing detailed information about the structure of the delay between the origin of the event and the variable of interest.

One interesting feature of a delay event study is the fact that we do not need to specify anything about the delay structure in order to run the test. Instead, by combining Equation (1) with the theoretical knowledge of delays our literature has developed, results in testable predictions about how the parameters we estimate during the delay event study will change based on the characteristics of the delay. Specifically, we expect that:

- Systems with very short delays should see $\beta_0$ rise to a significant fraction of its maximum slope $\beta_{max}$.
- Systems with significant first-order delays should see $\beta_0$ smaller than $\beta_{max}$, and a pattern where each subsequent $\beta$ increases at a decreasing rate, at least roughly.
- Systems with significant higher-order delays will also have $\beta_0$ much smaller than $\beta_{max}$, but will show more evidence of inflection, that is the $\beta$ will start off accelerating their rate of increase, before eventually slowing as they approach $\beta_{max}$.

The argument for the first expectation is straightforward. If the delay between the event and the response is short, then the very first $\beta$ we estimate will already exhibit almost all the eventual response. The other two expectations are derived from the response of different information delays to a step input. An excellent illustration of this behavior is Sterman's [4] Figure 11–14 from his page 434, which we recreate below as our Figure 1.

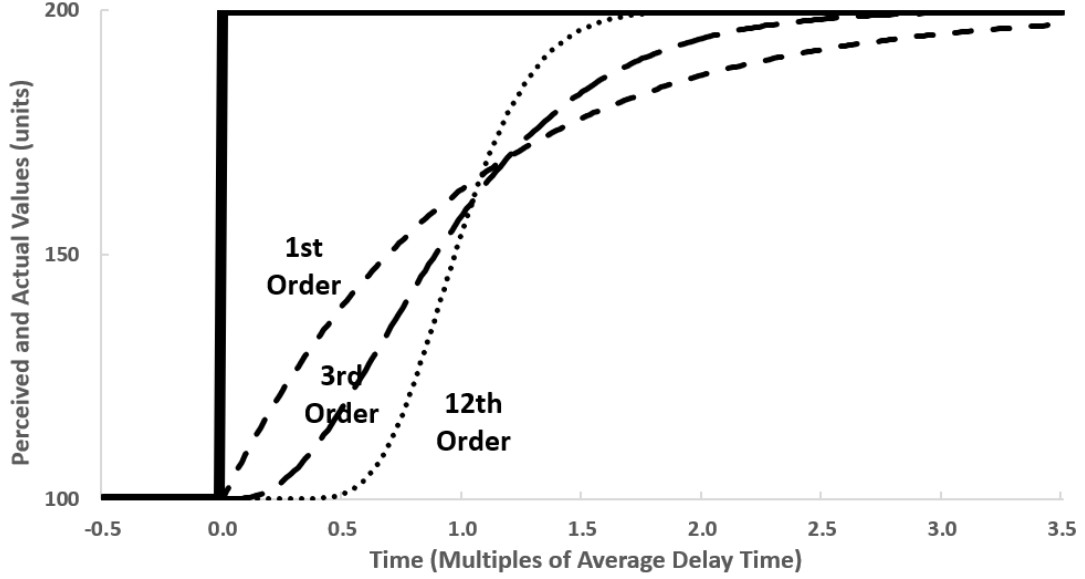

**Figure 1.** The Response of Higher Order Delays to a Step Input.

This step response is exactly what the $\beta$ we estimate during the delay event study are attempting to capture. The presence of absence of inflection in the $\beta$ estimates will indicate whether the delay is plausibly first order, or potentially higher-order, and the rapidity of this inflection can give a further indication of the order of the delay for very high order delays. While we cannot determine the average delay time precisely using this method, the delay order estimate combined with the number of samples it takes for the response to reach a large fraction of its maximum can give the modeler a rough idea of the average delay time by comparison with this chart.

---

[5]　This approach should work whether you expect the response to the event to be sustained like a step or transient like a pulse. The difference is that we would expect $D_\tau$ to move back towards zero if the response is transient.

That is not the limit of the analysis we can perform using a delay event study. If two or more variables of interest are in the expected path of the event, and if there is some uncertainty around which variable will respond first, then comparing the results of two different event studies is also potentially informative. In most cases though, direct comparison of the $\beta_0-\beta_T$ estimates between two delay event studies should only be done after scaling each set by their $\beta_{max}$, since in general there is no obvious reason why any two variables should change by the same percentage in response to some event.

Compare this process to the most common alternative, which is delay parameterization through partial model calibration. That approach would begin by tasking the modeler with designing a model of the causal pathway between the variables of interest. As part of that effort, the modeler would have to decide what the order of that delay might be, and what other structures or exogenous influences might be present. Then, once this notional model was completed, partial model calibration would drive the model's input with time series data and task the modeler with changing the delay structures and parameters to see what delay order, average delay time, and outside impacts best combine to fit the observed data [3].

While that approach is certainly well indicated for situations where there are only data on a single instance of a system, partial model calibration approaches the process of delay parameterization from a fundamentally different perspective than the delay event study does. Delay event studies are an important addition to the literature because they take the responsibility for deciding on possible delay orders and exogenous impacts away from the modeler, at least initially. By leveraging data on multiple instances of a system the delay event study can isolate the delay response while controlling for sources of exogenous noise. This provides direct guidance to modelers about the delay order and time constant even when there is little direct evidence on what those values should be *a priori*.

## 3. The Adequacy Era and School Funding

Starting in the late 1960s, legal theorists began arguing that the Constitution's Equal Protection Clause required that all students receive the same quality education. Despite losing a famous case at the federal level in San Antonio Independent School District v. Rodriguez (1973), many of these theorists, activists, and presumably disappointed parents took their arguments to the states, which often have specific clauses guaranteeing quality education in their constitutions.

The first state to face such a lawsuit was California [16] and that lawsuit's success helped to kick start a series of structural reforms in school financing. However, these cases focused on equalizing funding across districts rather than improving actual outcomes. Starting with a case in Kentucky in 1989, lawsuits began to focus instead on what they called "adequacy", which requires that school funding work to alleviate inequalities in the abilities or resources of incoming students [17]. Some went so far as to argue that this meant schools had to spend more in poorer districts in order to chase equality of outcomes rather than just equality of inputs. This "adequacy" era lead to similar cases being won in 28 states [18].

This series of court orders have been used by economists as quasi-experiments, since the states were forced to make changes in school funding that they otherwise would not have undertaken. The standard isolation technique in those studies involves examining student performance around the time of the court mandated changes in the hope that this will remove the influence of the political processes or selection issues that typically make testing the relationship between student performance and school funding troublesome[6]. This boils down to comparing the outcomes of poor students in adequacy states following a court order with a control group consisting of the outcomes of poor students in other states and the outcomes of poor students in adequacy states prior to a court order.

---

[6]  e.g., that richer districts might have students who are better prepared because of their home situation leading researchers to overestimate the relationship between funding and performance, or that some students with disabilities receive additional funding as part of a quasi-Rawlsian effort to level outcomes, leading researchers to underestimate the marginal effect of increased funding on average performance. For a detailed analysis of the determinants of district level spending, see [19].

Only poor students are expected to see any improvement because only those districts were "inadequate" to begin with, and so only those districts saw increased funding. Most recent papers accomplish this by running a separate regression for each quartile of districts grouped by median income prior to the adequacy era.

There is nothing inherently wrong with the mathematics of that approach, but this research design implicitly assumes that spending changes happened quickly following the court orders, since authors typically look for changes to outcomes starting immediately after the cases in question are decided [18]. A few recent papers do test to make sure that school spending actually changes in response to the court orders [9], but perhaps because of how much time has passed since the adequacy rulings, this literature is broadly uninterested in developing a careful understanding the characteristics of the delay between the court orders and spending changes.

There are ample theoretical reasons to expect such a delay, although its precise structure is less certain. Most school districts derive funding through a combination of the taxes and fees that they directly levy, and revenue sent to them by counties and states that levy their own taxes or impose other fees. The mixture of intergovernmental revenue from the state or county and each district's own source revenue from local taxpayers varies considerably between states [17].

Since the state, not any specific school district, is the typical plaintiff in these cases, it is immediately obvious that the causal pathway from court order to spending change will move through both a state budgetary process and a district level one. These processes take time. It is certainly possible that school district leaders might hear about a court order and work to preempt state budget makers by increasing spending before intergovernmental revenue arrives to cover it, but decades of research into budget making in the public sector suggest that a much more pessimistic budgetary attitude [20,21] is probably more likely.

## 4. Competing Delay Structures

In the context of school spending, we propose four models that our delay event study will attempt to distinguish between: A first-order delay with a short adjustment process, a first-order delay with a long adjustment process, a higher-order information delay driven only by compliance with the courts, and a higher-order information delay driven by both compliance and budgetary pessimism.

The implicit mental model of most economics research in this area is also a very simple model. Courts order changes in school spending, and schools respond quickly to that request. This can be easily represented by modelling the relationship with a simple causal link, as shown in Figure 2.

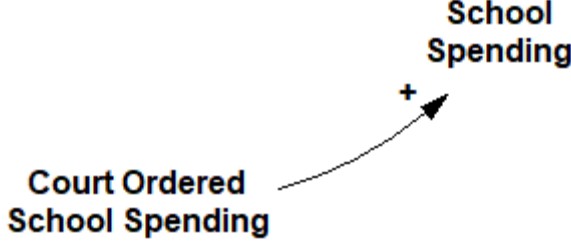

**Figure 2.** An illustration of the implicit mental model in adequacy research.

From the standpoint of a delay event study, this model is roughly equivalent to a model with a first-order delay where that delay is shorter than the sampling interval in the data. If $X$ is per student spending, then applying Equation (1) to a world that closely conforms with this mental model will result in a $\beta_0$ that is very close to the maximum $\beta_{max}$ with relatively similar values for all other $\beta_\tau$. Put another way, this model expects that when the court orders school spending to increase schools essentially instantly step up spending in response.

A simple extension of that model would be one where school spending responds to the court order with a significant delay. Figure 3 shows this first-order delay model.

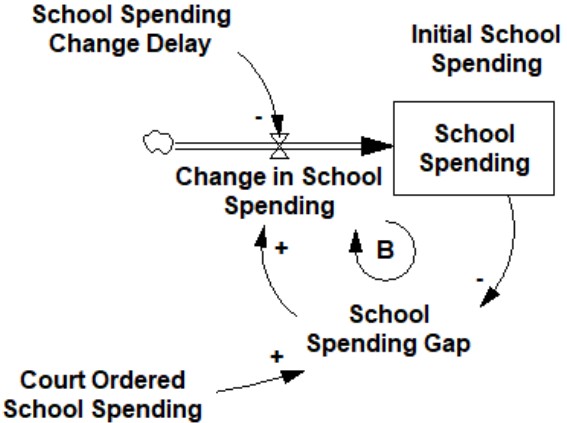

**Figure 3.** A first-order delay with a longer adjustment time.

This model is equivalent to a very simple delay differential equation:

$$\frac{dS}{dt} = \left(\frac{S_C - S}{\tau}\right) \tag{2}$$

where $S$ is the level of school spending, $S_c$ is the court ordered level of school spending, and $\tau$ is the adjustment time of the delay. We would expect a world that conformed to this model to have a relatively large gap between $\beta_0$ and $\beta_{max}$, but for that gap to close as the $\beta$ approached $\beta_{max}$ in a roughly exponential fashion. In other words, we would expect the $\beta$ to look like the response of a first-order delay to a step input, as illustrated in Figure 1.

A more realistic model of this process might employ a series of information delays to describe the actual path that any court ordered spending change would take before it can influence spending for a specific district. A parsimonious description might conceive of three intermediate steps in this process. First state legislators must learn about the court order and update their expectations for school spending, then a budget with the appropriate spending changes must be passed at the state level, and only then can actual spending changes be implemented at the district level.

Essentially this model for spending changes would be:

$$\frac{dS}{dt} = \left(\frac{S_B - S}{\tau_1}\right) \tag{3}$$

$$\frac{dS_B}{dt} = \left(\frac{S_L - S_B}{\tau_2}\right) \tag{4}$$

$$\frac{dS_L}{dt} = \left(\frac{S_L - S_C}{\tau_3}\right) \tag{5}$$

where $S_L$ represents the spending level legislators want to appropriate, and $S_B$ represents the spending level that legislatures budget for. Each stage of this cascading process might potentially have its own characteristic delay time $\tau_N$.

This model takes the form of a third-order information delay, as shown in Figure 4.

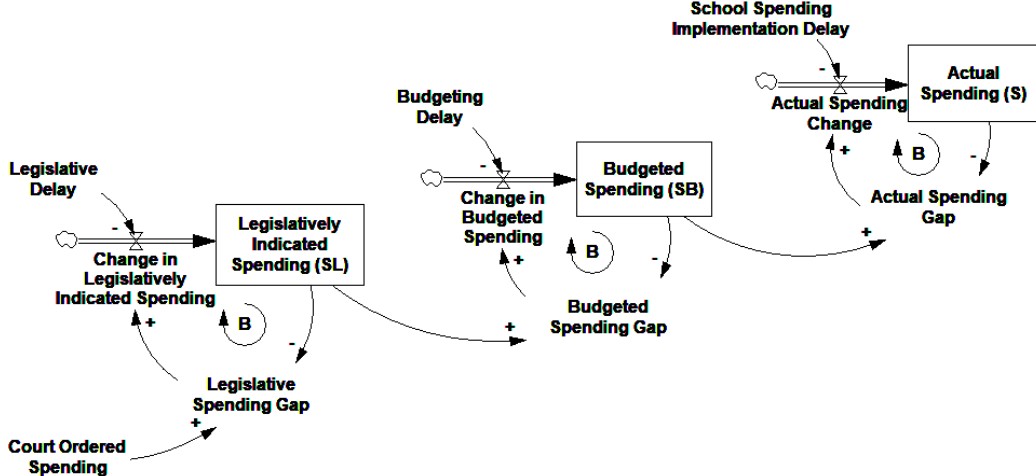

**Figure 4.** Implementation of the third-order delay model.

Applying Equation (1) to data from a world that closely conforms to this system will result in a set of $\beta_\tau$ that increase in an s-shaped fashion up to their maximum value, which is the characteristic path of a higher-order delay following a step increase.

The third-order model shown in Figure 4 is more operational than the two first-order models, but it may still be ignoring one potential source of delay: The reluctance of any prudent financial manager to spend more money than they have. There is ample evidence that public sector budget makers create pessimistic forecasts in some[7] settings [21,22]. If school districts produce pessimistic budget forecasts, then we might expect their behavior following a court order to more closely follow a slightly different model, which we illustrate in Figure 5.

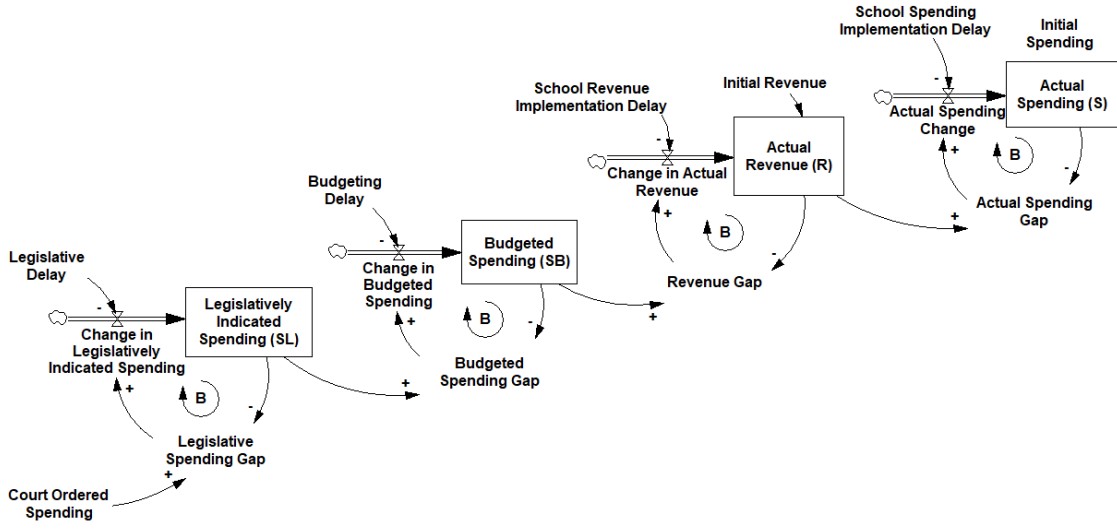

**Figure 5.** Higher-order delay model where revenue drives spending.

It is important to note that without separate time series for both spending and revenue a delay event study could not tell the difference between the model in Figure 5 and the model in Figure 4. Both would produce an s-shaped pattern in the $\beta_\tau$ for school spending. We can overcome this limitation by running two separate tests: One that uses revenue as its outcome variable and another which uses

---

7    While there is also some evidence that public sector forecasts can be overly optimistic our tests are already capable of
     detecting that behavior by rejecting a high order model.

spending. If school districts wait for new revenue to arrive before they decide to change spending, then the pattern of $\beta_\tau$ produced in tests using revenue will increase faster than the pattern of $\beta_\tau$ produced in tests using spending. Formally, we would expect each element of $\beta_\tau$ for the revenue regression to be greater than or equal to the corresponding element of $\beta_\tau$ for the spending regression, at least when expressed as a percentage of their two maximum values revenue $\beta_{max}$ and spending $\beta_{max}$.

## 5. Data and Empirical Tests

There are ample data available to explore school funding questions. We use data from the individual unit files of the annual surveys of government finance provided by the US Census, as organized by Pierson et al. [23]. This source provides 529,063 district-years of financial data, including a subsample of all US school districts from 1970 through 2014, and the population of school districts in years ending in a 2 or a 7, when the Census of Governments is run. The size of each subsample is considerable, often covering more than 90% of all school districts in every year. The one exception is the period between 1993 and 1996 when only around 3000 of the 16,895 school districts in the panel were sampled.

The data include detailed information on both revenue and expenses. For our tests, we focus on two broad categorizations: Current expenses, which ignore lumpy capital investments, and total revenue, which counts both direct aid to the school district as well as revenue the district raises directly. These two variables are expressed in thousands of real, 1983 dollars[8], per student enrolled in the district, and are transformed using natural logs in order to fit Equation (1).

Data for our treatment events come from Lafortune et al. [18], specifically we take their court order dates from Appendix Table A1 of their paper. We generate several lagged treatment variables, which take a value of one for each district-year data point a specific number of years following the court order. Observations 11 or more years after a court order are lumped together into a single indicator. States with multiple court orders are only considered as treated by the first one.

We require observations to have non-missing and non-zero enrollment (17,438 observations lost) as well as non-missing total expenditure (3956 observations lost) in order to be included in our panel. Observations with missing data for any one test were excluded from that test but could remain in our other tests. We do not require a district to be sampled a minimum number of times in order to be included in the panel, since smaller districts are much less likely to be sampled by the census [23]. The final sample contains 454,887 observations across 12,498 districts.

Our first delay event study is designed to detect the response of school district current expenses $C$ to the adequacy era court orders. This test is performed following Equation (6):

$$Ln(C_{dt}) = \alpha + \sum_{\tau=0}^{11} \beta_\tau D_\tau + \delta_d + \delta_t + \delta_g + \theta_1 S_{dt} + \varepsilon_{st} \tag{6}$$

where $d$ indexes school districts, $t$ indexes time, and $\tau$ is the time when each school district experiences a court order. $\delta_d$ is a fixed effect for each school district, $\delta_t$ is a fixed effect for each year of the data, and $\delta_g$ is a unit specific time trend. We chose to control for only a single variable, the enrollment size S of each district. This is an important variable for us to control for because we expect differently sized school districts to respond differently *on a percentage basis* to court ordered spending changes. Larger districts spend less per student because of the inherent economies of scale available in education [24]. As a result, the percentage change in per-student spending that is needed to achieve adequacy will naturally be smaller the more students there are enrolled in a district.

---

8    It is not strictly necessary to transform nominal dollars to real dollars since the time fixed effects are annual and so will absorb any inflation in the annual reported values.

Given the fact that adequacy era spending changes should be concentrated in the poorest districts, we follow the approach of most papers in economics [9], and restrict our tests to only include school districts that are in the lowest quartile of per capita median income, as measured by the 1979 census. Unreported tests performed using controls for previous relative funding levels, as well as tests that used different definitions for median income produced similar results.

In order to differentiate between a system where school spending is driven primarily by state budgets and a system where school spending experiences an additional delay due to pessimistic budget forecasts, we also test Equation (7):

$$Ln(R_{dt}) = \alpha + \sum_{\tau=0}^{11} \beta_\tau D_\tau + \delta_d + \delta_t + \delta_g + \theta_1 S_{dt} + \varepsilon_{st} \tag{7}$$

The only difference between Equations (6) and (7) is that Equation (7) models per student total revenue $R$ rather than current expenses $C$. If the scaled $\beta_\tau$ from Equation (7) react more quickly than the scaled $\beta_\tau$ from Equation (6), we can infer that school revenues increase before school expenses do, at least on average, and so pessimistic budget forecasts may be causing an additional delay following state level changes in education spending.

## 6. Results

The slope estimates from regression tests of Equations (6) and (7) are shown in Table 1. While the statistical significance of these estimates is not of primary importance, all the reported standard errors are clustered by state [25].

**Table 1.** Revenues and expenses after a court order.

|  | Current Expenses 6 | Revenue 7 |
|---|---|---|
| Size ($S_{dt}$) | −0.431 *** | −0.442 *** |
|  | (0.0256) | (0.0296) |
| $\beta_0$—year of court order | 0.00307 | 0.00308 |
|  | (0.00906) | (0.0127) |
| $\beta_1$—one year after | 0.0124 | 0.0558 |
|  | (0.0143) | (0.0457) |
| $\beta_2$—two years after | 0.0121 | 0.00913 |
|  | (0.0116) | (0.0283) |
| $\beta_3$—three years after | 0.0313 ** | 0.0434 |
|  | (0.0132) | (0.0257) |
| $\beta_4$—four years after | 0.0267 * | 0.0415 ** |
|  | (0.0140) | (0.0155) |
| $\beta_5$—five years after | 0.0363 ** | 0.0427 ** |
|  | (0.0172) | (0.0175) |
| $\beta_6$—six years after | 0.0509 *** | 0.0446 ** |
|  | (0.0171) | (0.0185) |
| $\beta_7$—seven years after | 0.0586 *** | 0.0484 *** |
|  | (0.0160) | (0.0178) |
| $\beta_8$—eight years after | 0.0584 *** | 0.0433 ** |
|  | (0.0175) | (0.0201) |
| $\beta_9$—nine years after | 0.0643 *** | 0.0392 ** |
|  | (0.0167) | (0.0190) |
| $\beta_{10}$—ten years after | 0.0552 *** | 0.0102 |
|  | (0.0198) | (0.0259) |
| $\beta_{11}$—more than ten years after | 0.0431 | 0.00539 |
|  | (0.0310) | (0.0372) |
| Observations | 113,665 | 113,664 |
| Number of districts | 3356 | 3356 |
| R-squared | 0.926 | 0.900 |

Robust standard errors in parentheses. *** $p < 0.01$, ** $p < 0.05$, * $p < 0.1$

These estimates make it immediately apparent that the naïve model that expects school spending to change quickly in response to a court order is not a good description of reality. The maximum change in school spending we estimate, $\beta_{max}$, is 0.0643. The initial response to the court order, $\beta_0$, is 0.00307, which only represents around 5% of the eventual maximum.

The first-order information delay with a long adjustment time fits the pattern of $\beta_\tau$ we estimate significantly better than the naïve model, since the maximum response of expenses is not estimated until nine years after the court order. Differentiating between the first-order delay and a higher-order delay however is much less clear. Indeed, the graph of the response of expenses shown in Figure 6 seems to almost straddle the line between a clearly exponential path and a clearly S-shaped one. We could settle for a compromise and propose a second-order delay for our model, but we can also approach the question using a more quantitative method.

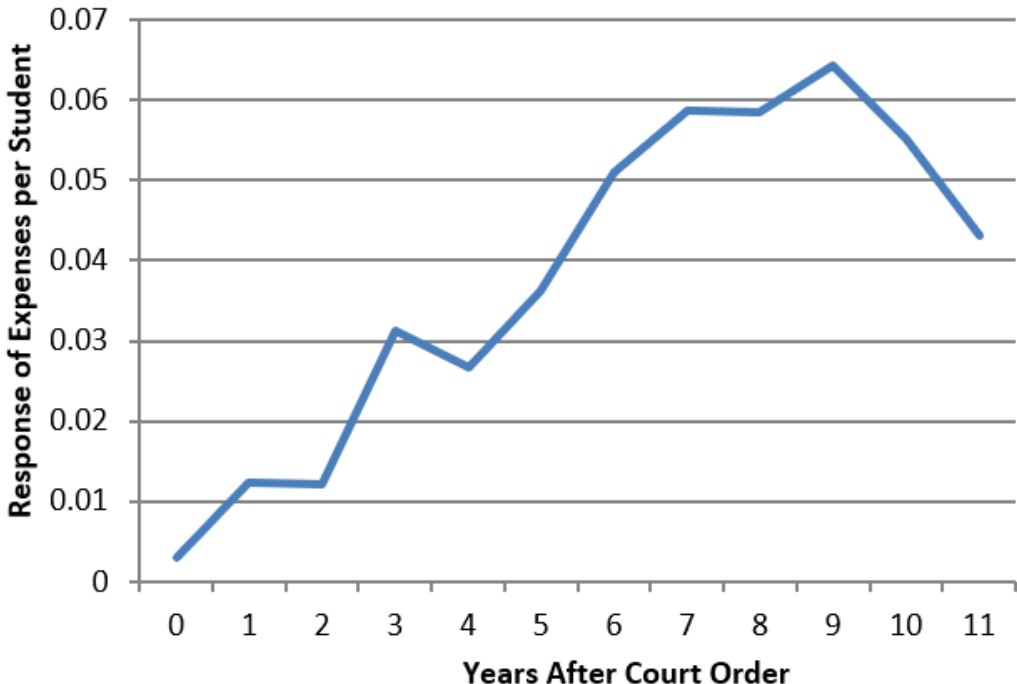

**Figure 6.** A graph of the estimated response of expenses following a court order.

In order to quantify whether the $\beta$ we estimate show evidence of inflection, and therefore whether the system we are testing data for is more like a first-order delay or more like a higher-order delay, Table 2 expresses each $\beta$ estimate for the response of current expenses as a fraction of the maximum value $\beta_{max}$ and reports their first differences.

As Figure 1 indicates, the response of a first-order delay to a step input will be to exponentially adjust to its maximum, whereas a higher-order delay will logistically adjust to its maximum. Thus, we would expect a first-order system to have $\beta$ estimates from a delay event study that increase rapidly at first and then slowly decrease their rate of increase. A higher-order delay would have three phases to its response: an initial rapid increase, followed by a sustained pace of increase, and a final phase where the rate of increase slows rapidly. The higher a delay order is, the faster the transition from essentially no response to a rapidly increasing response, and the faster the transition from a rapidly increasing response to the unchanging maximum response.

**Table 2.** The scaled response of school expenses.

| Expenses | Scaled Response $\beta_\tau/\beta_{max}$ | First Differences |
|---|---|---|
| $\beta_0$—year of court order | 4.8% | |
| $\beta_1$—one year after | 19.3% | 14.5% |
| $\beta_2$—two years after | 18.8% | −0.5% |
| $\beta_3$—three years after | 48.7% | 29.9% |
| $\beta_4$—four years after | 41.5% | −7.2% |
| $\beta_5$—five years after | 56.5% | 14.9% |
| $\beta_6$—six years after | 79.2% | 22.7% |
| $\beta_7$—seven years after | 91.1% | 12.0% |
| $\beta_8$—eight years after | 90.8% | −0.3% |
| $\beta_9$—nine years after | 100.0% | 9.2% |
| $\beta_{10}$—ten years after | 85.8% | −14.2% |
| $\beta_{11}$—more than ten years after | 67.0% | −18.8% |

We certainly cannot use Table 2 to completely settle the argument over order of this delay, since the rate of growth in the scaled responses is far from consistent. That said, the column of first differences we calculate seem to fit the pattern of a higher-order delay where growth speeds up, holds steady, and then slows down again reasonably well. They certainly do not fit the pattern of a first-order delay where growth starts fast and then monotonically slows down over time. In our estimation then, these results suggest that a higher-order delay is probably the better description of the system producing these results than a first-order delay would be.

The graph in Figure 7 reports scaled response estimates for expenses on the same axis as the scaled response estimates for revenue.

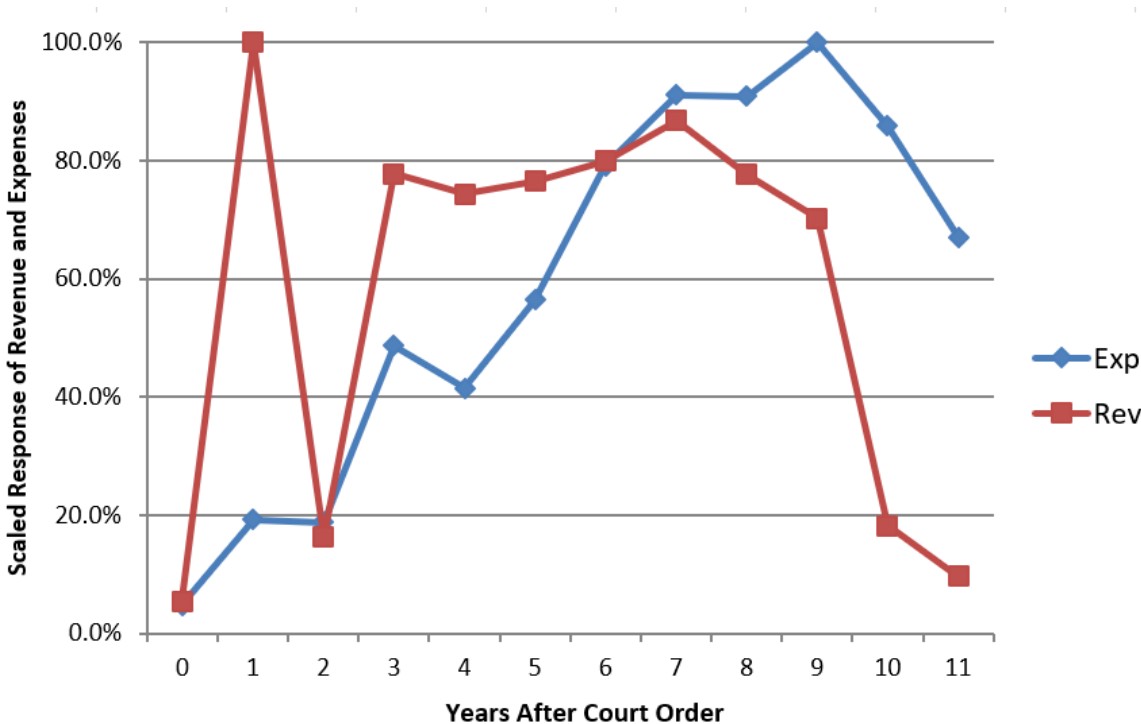

**Figure 7.** Delay event study response estimates as a percent of their maximum.

The response we estimate for revenue following a court order is considerably different from the response we estimate for expenses. Revenue jumps to its maximum value only one year following a court order, but then returns to a much lower level, before jumping up to close to 80% of its maximum and slowly declining. Two things are made clear by this result. First, revenue reacts much more

quickly to a court order than expenses do, and second, revenue changes are much less likely to persist over time.

This finding supports the notion that pessimistic revenue forecasts by school district managers delayed the implementation of school spending changes in response to the adequacy era court orders, but it also provides an excellent behavioral excuse for those pessimistic forecasts. If new revenue that has been allocated might suddenly disappear next year, or even over a decade, most reasonable managers should think twice about committing to any new teacher contracts or student services. Also, the fact that the revenue estimates fall off much more quickly than the expense estimates following year 7 further validates the idea that school expenses only adjust to revenue with a significant delay.

## 7. Conclusions

These results indicate that a higher-order delay structure that includes both the relatively quick adjustment of revenue to the court order and a slower adjustment of expenses to changes in revenue is the most likely structure for this system. That said, these results are not perfect. We offer them here because we believe that it is important to begin this discussion in the hope that others will supplement it with delay event studies in other contexts.

Overall, delay event studies enable us to build significant insight into the structure of a complex social system without any simulation modeling. While it is possible that parameterizing this delay inside of a fully developed model would produce different results, one of these approaches does not preclude the other. This relatively simple empirical test of the structure of a delay is a step forward in the literature on delay estimation. We offer our exposition in part to encourage others to pursue similar tests so that we can work together as a community to better understand when delay event studies are appropriate tools to use to better understand the structure of complex systems.

**Author Contributions:** Conceptualization, K.P. and J.C.T.; Data curation, K.P.; Methodology, J.C.T.; Writing–original draft, K.P. and J.C.T.; Writing–review & editing, K.P. and J.C.T. All authors have read and agreed to the published version of the manuscript.

**Funding:** This research received no external funding.

**Conflicts of Interest:** The authors declare that they have no conflicts of interest.

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
