# Peer review of "Leveraging Quasi-Experimental Methods to Estimate Model Structure: Understanding School Funding Changes in Response to Court Orders"

_systems, doi:10.3390/systems8030025_

Round 1
Reviewer 1 Report
This paper is well written and particularly well positioned in terms of its significance and relevance to system dynamics modelers.
I would find it useful if the authors were to provide a bit more example on the alternative approaches, such as "parameterizing this delay inside of a fully developed model". Comparing their results to such an exercise would be very interesting, but I understand it may be out of scope for the paper. For readers less familiar with system dynamics, perhaps they could at least explain how that would work, so that we can see how much simpler this approach is.
I also found it hard to follow the discussion around table 2, which is supposed to settle the debate between first order delays and higher order delays. Is this just the slopes of Figure 5, and if so, why can that settle the debate when Figure 5 could not? If I have misunderstood, please add some clarity here.
In general, the paper is well written and I appreciate the goal to simplify parameterization of system dynamics models.
Author Response
Thank you very much for this helpful review. Our responses are detailed in the attached document.

Reviewer 2 Report
I liked the intent and execution of the paper very much. I am convinced that this is a good method for parameterizing delays in system dynamics models.
I have only two minor suggestions for improvement:
- Early in the paper, the authors mention the "adequacy era" without explanation. They explain it in detail, and very well, later, but the reader might benefit from a short explanation at the first mention.
- I followed the authors' arguments about shapes of first- and third-order delays, but it might have been helpful to include some "standard" images of those types of delays, perhaps from Sterman's book. That way, the reader would more easily see the similarities between the study's results and the shapes of the classical delay types.
Author Response

(The authors gave the same response as above.)
